# Prediction of Bioactive Peptides from *Chlorella sorokiniana* Proteins Using Proteomic Techniques in Combination with Bioinformatics Analyses

**DOI:** 10.3390/ijms20071786

**Published:** 2019-04-11

**Authors:** Lhumen A. Tejano, Jose P. Peralta, Encarnacion Emilia S. Yap, Fenny Crista A. Panjaitan, Yu-Wei Chang

**Affiliations:** 1Institute of Fish Processing Technology, College of Fisheries and Ocean Sciences, University of the Philippines Visayas, Miagao 5023, Iloilo, Philippines; lhumentejano@gmail.com (L.A.T.); f153mentor@yahoo.com (J.P.P.); esyap@up.edu.ph (E.E.S.Y.); 2Department of Food Science, National Taiwan Ocean University, Keelung 202, Taiwan; fennycap@gmail.com

**Keywords:** *Chlorella sorokiniana*, in silico, BIOPEP-UWM database, proteomics, bioactive peptides, nano liquid chromatography tandem mass spectrometry (nanoLC–nanoESI MS/MS)

## Abstract

*Chlorella* is one of the most nutritionally important microalgae with high protein content and can be a good source of potential bioactive peptides. In the current study, isolated proteins from *Chlorella sorokiniana* were subjected to in silico analysis to predict potential peptides with biological activities. Molecular characteristics of proteins were analyzed by sodium dodecyl sulfate polyacrylamide gel electrophoresis (SDS-PAGE) and proteomics techniques. A total of eight proteins were identified by proteomics techniques from 10 protein bands of the SDS-PAGE. The predictive result by BIOPEP’s profile of bioactive peptides tools suggested that proteins of *C. sorokiniana* have the highest number of dipeptidyl peptidase-IV (DPP IV) inhibitors, with high occurrence of other bioactive peptides such as angiotensin-I converting enzyme (ACE) inhibitor, glucose uptake stimulant, antioxidant, regulating, anti-amnestic and antithrombotic peptides. In silico analysis of enzymatic hydrolysis revealed that pepsin (pH > 2), bromelain and papain were proteases that can release relatively larger quantity of bioactive peptides. In addition, combinations of different enzymes in hydrolysis were observed to dispense higher numbers of bioactive peptides from proteins compared to using individual proteases. Results suggest the potential of protein isolated from *C. sorokiniana* could be a source of high value products with pharmaceutical and nutraceutical application potential.

## 1. Introduction

Microalgae are eukaryotic unicellular organisms that grow easily with inexpensive substrates. Therefore, they are considered to be economical and effective raw materials in industry [1]. Many studies have been conducted to utilize microalgae as useful products. Most of them majorly focused on the potential of microalgae for biofuel production due to their lipid content and abundant availability [1,2]. However, due to the increasing population and demand for protein, there is a call to further utilize microalgae as protein sources to shift away from animal proteins.

Microalgae have been known as one of many promising alternative plants for proteins as they offer up to 50% (*w/w*) of protein [3] with a well-balanced amino acid profile required for the nutrition of human beings [1,4]. Several studies have reported the biological activities of various microalgae protein hydrolysates, including immunostimulant and antitumor activities from *C. sorokiniana* [5,6], angiotensin-I converting enzyme (ACE) inhibitory and hypotensive activities from *Chlorella vulgaris* [7,8] and *Nannochloropsis oculata* [9], antioxidant effects from *Navicula incerta* [10] and *Chlorella ellipsoidea* [11], anti-inflammatory effects from *Spirulina maxima* [12] and antibacterial property from *Spirulina platensis* [13]. From *Chlorella sorokiniana*, Lin et al. [14] were able to purify and identify four active peptides with high ACE inhibitory effects. A variety of other compounds were also detected from microalgae which show certain usefulness for human and animals. Morgese et al. [15] previously reported the beneficial effect of omega (ω)-3 and ω-6 polyunsaturated fatty acids (PUFAs) from *Chlorella sorokiniana* on the emotional, cognitive and social behavior in rats. Talero et al. [16] also reviewed bioactive compounds of microalgae that give the chemopreventive effect on chronic inflammation and cancer. Corresponding to those findings, the exploration of biological activities from microalgae has gained significant attention with regard to its health-promoting properties related to bioactive compounds. Therefore, further observations are still needed to identify more bioactive peptides from *C. sorokiniana*, including dipeptidyl peptidase-IV (DPP IV) inhibitory peptides, ACE-inhibitory peptides and antioxidant peptides.

With the advancements in protein analysis, techniques for protein identification and predictive analysis of potential bioactive peptides have been established. Proteomics techniques have been widely used to analyze proteins presented in protein sample [17]. Moreover, mass spectrometry (MS)-based proteomics have been successfully applied in researches to identify protein from complex materials, including protein characterization of chickpea and oat seeds [18], fish authentication [19], species identification of spoilage and pathogenic bacteria [20], and protein characterization of tilapia processing co-products [21]. Once the sequences of protein are obtained, a bioinformatics tool such as the BIOPEP-UWM database is used to predict bioactive peptides composed in protein sequences [22,23]. Bioinformatics, also known as in silico technique, is a computational method used to estimate bioactive peptides from the known protein sequences [24]. It also allows performing the simulation of enzymatic hydrolysis using proteases to predict bioactive peptides theoretically released from the intact protein sequences [25,26,27].

The application of proteomics coupled with BIOPEP-UWM will be able to deliver a rapid method to identify and characterize proteins. This technique will reduce cost and time regarding the prediction of the potential bioactive peptides. Thus, this study generally aimed to characterize the isolated proteins from *C. sorokiniana* using in-gel digestion and proteomics techniques. Furthermore, the BIOPEP-UWM database tool was used to predict the potential bioactive peptides derived from the identified proteins of *C. sorokiniana*.

## 2. Results and Discussion

### 2.1. Identified Proteins of C. sorokiniana

The protein content of *C. sorokiniana* isolates was 65.08 ± 0.88% with a yield of 4.40% (w/w initial biomass dry basis). A total of the 10 distinct protein bands of *C. sorokiniana* proteins were observed in 12% acrylamide gel (Figure 1). These labeled protein bands (A–J) were used for the in-gel digestion and subsequently analyzed using nanoLC–nanoESI MS/MS. The molecular weights (MWs) of the proteins estimated by SDS-PAGE were 109.02 (A), 72.08 (B), 54.15 (C), 45.72 (D), 38.33 (E), 29.04 (F), 24.96 (G), 21.13 (H), 16.59 (I), and 7.82 (J). Eight protein hits were discovered from the selected bands in the NCBI database namely, chloroplast rubisco activase, 50S ribosomal protein L7/L12 (chloroplast), phosphoglycerate kinase, Fe-superoxide dismutase, heat shock protein 70, ATP synthase subunit beta (chloroplast), elongation factor 2, partial, and V-type H+ ATPase subunit A, partial. The protein hits, accession number from the NCBI database, length of amino acid (AA), and molecular weight of the band reported in NCBI database and estimated by SDS-PAGE were presented in Table 1. The estimated MWs by SDS-PAGE were comparable to the theoretical molecular weights reported in NCBI database except for elongation factor 2, partial and V-type H+ ATPase Subunit A, partial.

*C. sorokiniana* is a freshwater green algae species with high protein content [28]. This species of genus *Chlorella* was originally called as *C. pyrenoidosa* [29]. The NCBI database revealed a total of 20,925 proteins from the *C. sorokiniana*. Most of them are enzymes responsible for various cell functions. According to the nanoLC–nanoESI MS/MS data, eight protein hits from the NCBI database corresponded to the proteins of *C. sorokiniana*. Phosphoglycerate kinase (*Auxenochlorella pyrenoidosa*, NCBI accession number: AKP17751.1) was detected in all selected protein bands in the SDS-PAGE. Watson et al. [30] stated that phosphoglycerate kinase derived from various protein sources are all monomers with MWs around 45 kDa. Moreover, their amino acid composition and catalytic functions are similar [31]. The reference molecular weight for phosphoglycerate kinase from NCBI database was 49.13 kDa, thus phosphoglycerate kinase found in band D (45.72 kDa) was used for further analyses. The possibility of having the same single protein in different bands is high since proteins are denatured and separated in SDS-PAGE. In addition, protein hits discovered by the Mascot database were identified based on the matched tryptic peptides detected by mass spectrometry.

### 2.2. Identified Tryptic Peptides from C. sorokiniana Proteins

Tryptic peptides derived from the identified proteins of *C. sorokiniana* by in-gel digestion were evaluated by nanoLC–nanoESI MS/MS analysis. Those tryptic peptides identified through mass spectrometry were peptides matching with protein hits in the Mascot database. Tryptic peptides are generated by trypsin through in-gel digestion process which is part of proteomics technique. The result of the MS/MS ion search for the tryptic digests revealed that all the tryptic peptides from the identified proteins were doubly and triply charged. Figure 2 and Figure 3 present the representative spectra of the doubly and triply charged peptides from the identified proteins of *C. sorokiniana* by proteomics analysis.

Figure 2 illustrates the doubly charged tryptic peptide of *C. sorokiniana* of protein band D (Figure 1), with observed signal m/z 1053.01 marked in red box representing a doubly charged peptide (with adjacent signal difference of 0.50, insert A), and nanoLC–nanoESI MS/MS fragmentation spectra of NFNNIEDGFYISPAFLDK found in chloroplast rubisco activase (NCBI accession no. AEL29575.1) represented in insert B. Figure 3 illustrates the triply charged tryptic peptide also found in the same identified protein of the same protein band D. The observed signal was m/z 618.66, also marked in red box demonstrating a triply charged peptide (with adjacent signal difference of 0.33, insert A), and nanoLC–nanoESI MS/MS fragmentation spectra of tryptic peptide LVDAFPGQSIDFFGALR found in chloroplast rubisco activase (NCBI accession no. AEL29575.1) is illustrated in insert B.

In the identification of proteins by proteomics analysis, trypsin is usually used to digest proteins in the gel [32]. Trypsin hydrolyzes protein specifically at the C-terminus of the carboxyl side of the amino acids arginine or lysine, but poorly when lysine and arginine are followed by proline. With this perspective, the tryptic peptides are either doubly or triply charged in ESI since the amino terminal residues are basic which explains the result of the MS/MS ion search [33].

### 2.3. Potential Bioactive Peptides from Identified Proteins in C. sorokiniana

Potential bioactive peptides presented in identified proteins of *C. sorokiniana* were investigated using the BIOPEP-UWM database. Amino acid sequences of six proteins, namely chloroplast rubisco activase, 50s ribosomal protein l7/l12 (chloroplast), phosphoglycerate kinase, Fe-superoxide dismutase, heat shock protein 70 and ATP synthase subunit beta (chloroplast), were chosen as they are found to be relatively abundant components of *C. sorokiniana* proteins found in SDS-PAGE based on the results. Moreover, they also corresponded to the estimated molecular weights in the NCBI database (Table 1). The profile of the potential bioactive peptides, their biological activities (ACE inhibitory, antioxidant, anti-amnestic, antithrombotic, stimulating, regulating, DPP IV inhibitory), and frequencies are summarized in Table 2. Results revealed that most of the potential bioactive peptides were dipeptides or tripeptides with multiple biological activities. The number of those bioactive peptides was identified based on the amino acid sequences which were predicted to become potential bioactive peptides. The BIOPEP database displays peptides with their bioactivities from inputted protein sequences corresponding to the information in the database.

Chloroplast rubisco activase (NCBI accession no. AEL29575.1) and phosphoglycerate kinase (NCBI accession no. AKP17751.1) were chosen to illustrate the profile of bioactive peptides within in the protein (Figure 4 and Figure 5) because they appeared to be relatively abundant and were found in almost all picked bands in SDS-PAGE. As shown in Figure 4, the molecular weights of the tryptic peptides corresponded to the identified tryptic peptide at amino acid positions 100–117, 132–144, 158–181, 187–145, 149–213, 255–273, 290–304, and 310–326 (matched tryptic peptides shown in red letters) in the chloroplast rubisco activase amino acid sequence. BIOPEP-UWM analysis results exhibited that potential bioactive peptides encrypted in chloroplast rubisco activase amino acid sequence were mostly DPP IV inhibitors (with 250 peptide fragments, marked with an orange line) and ACE inhibitors (with 187 peptide fragments, marked with a green line). Other bioactive peptides found were 18 antioxidant, 3 anti-amnestic, 2 antithrombotic, 18 stimulant and 3 regulatory peptides. Some bioactive peptides have multiple activities such as VPL, WG, LA, IR, PG, VY, and KP.

On the other hand, Figure 5 shows the molecular weights of the tryptic peptides in phosphoglycerate kinase which corresponds to the theoretical tryptic peptides at amino acid positions 232–244, 258–268, 285–298, 303–313, 383–411, and 436–465 (matched tryptic peptides shown in red letters). There were 297 DPP IV inhibitor, 224 ACE inhibitor, 23 antioxidant, 33 stimulant, 5 anti-amnetic, 4 antithrombotic, and 6 regulatory peptides embedded in the amino acid sequence of phosphoglycerate kinase. Moreover, the profiles of the bioactive peptides of 50s ribosomal protein l7/l12 (chloroplast), Fe-superoxide dismutase, heat shock protein 70 and ATP synthase subunit beta (chloroplast) also show the presence of the above mentioned bioactive peptides in these proteins, except that Fe-superoxide dismutase does not show anti-amnestic, antithrombotic, or regulating peptides. In all the proteins, DPP IV and ACE inhibitors were the most abundant bioactive peptides.

The amino acid composition and sequence of the proteins greatly determines the presence of these bioactive peptides. Results also revealed that most of the DPP IV peptides present in the identified proteins had proline (P), alanine (A), glycine (G), valine (V) and leucine (L) amino acid residues. DPP IV preferably cleaves dipeptides with proline and alanine residues at the N-terminal side of the peptide [34]. It also has relatively lower cleavage rates with serine, glycine, leucine, and valine [35,36]. Moreover, the presence of basic and hydrophobic amino acids at the N-terminal side of the peptides could enhance the cleavage susceptibility of the substrate [34,37]. DPP IV inhibitors were also reported from various protein sources by in silico approach including barley, canola, oat, soybean, wheat, quinoa, chicken egg, bovine milk, bovine meat, pig, tuna, Atlantic salmon, chum salmon, tilapia skin and frame, and palmaria palmate [21,27,38,39,40]. On the other hand, the abundance of ACE inhibitory peptides in the identified proteins might have also been influenced by the amino acid compositions of the proteins. The presence of amino acid residues such as phenylalanine (F), tyrosine (Y), tryptophan (W), or proline (P) in at the C-terminal side of the peptides have been reported to exhibit high potent ACE inhibitory activity [41,42,43,44,45]. The adjacent amino acid residue of proline can also influence the potency of the ACE inhibitor, which is usually enhanced by hydrophobic amino acids [46]. In silico analysis of different proteins revealed the abundance of ACE inhibitors embedded in various protein sequences [47,48,49]. In previous studies, some amino acid sequences of ACE inhibitory peptides from *C. sorokiniana* were discovered. Lin et al. [14] reported IC_50_ values of WV, VW, IW, and LW were 307.61, 0.58, 0.50, and 1.11 µM, respectively. Moreover, *C. sorokiniana* protein hydrolysates could reduce systolic and diastolic blood pressure at 20 and 21 mm Hg, respectively. Suetsuna and Chen [8] also mentioned several amino acid sequences generated potential antihypertensive activity through oral administration, such as IVVE (IC_50_: 315.3 μM), AFL (IC_50_: 63.8 μM), FAL (IC_50_: 26.3 μM), AEL (IC_50_: 57.1 μM), and VVPPA (IC_50_: 79.5 μM) from *C. vulgaris*; IAE (IC_50_: 34.7 μM), FAL, AEL, IAPG (IC_50_: 11.4 μM), and VAF (IC_50_: 35.8 μM) from *S. platensis*. Those findings showed that ACE inhibitory peptides predicted through in silico analysis obviously possessed potential antihypertensive activity through in vitro and in vivo analysis.

For many years now, in silico analysis has been successfully used to predict the potential application of various proteins as a source of bioactive peptides [22]. It provides sufficient information for determining the potential biological activity of proteins which is much faster than conventional methods [21]. The results of the in silico analysis by BIOPEP-UWM suggest the potential of *C. sorokiniana* proteins for pharmaceutical application as demonstrated by its bioactivities. These peptides in the intact proteins are inactive and need to be released in order to perform their functions [50]. The prediction of the potential bioactivities of the proteins after digestion by various proteases can be conducted by the BIOPEP-UWM database tool.

### 2.4. Prediction of Potential Bioactive Peptides after Protease Cleavage using BIOPEP-UWM Tool

Identified proteins such as chloroplast rubisco activase, phosphoglycerate kinase, Fe-superoxide dismutase, heat shock protein 70 and ATP synthase subunit beta (chloroplast) were further analyzed using the “enzyme action” tool in BIOPEP-UWM database; these proteins showed the most numbers of bioactivities from their profiles of bioactive peptides (Table 2). Results of the 15 simulations of enzymatic hydrolysis for each protein sequence are presented in Table 3. The table shows the number of bioactive peptides with specific bioactivities after hydrolysis of the individual proteins by various proteases. The results revealed that DPP IV inhibitory peptides were observed to be dominantly produced from the selected proteins using different proteases. ACE inhibitory peptides were also released in relatively high numbers but lower than DPP IV. This information is in concurrence with the profile of the potential bioactive peptides from the proteins in Table 2. Bromelain, papain, ficin and pepsin (pH > 2) were the individual proteases that released the most diverse and large number peptides with certain biological activities from all the selected proteins. Meanwhile, trypsin had the lowest number of bioactive peptides release after in silico hydrolysis. Trypsin is the most commonly used enzyme in proteomics approach [32], however, in the in silico analysis, it did not release significant numbers of potential bioactive peptides. Nonetheless, based on the results, other single action enzymes could also release relatively high numbers of bioactive peptides. The use of a combination of enzymes in hydrolysis is also offered by the BIOPEP-UWM database. A combination of two to a maximum of three enzymes could be utilized in the hydrolysis simulation of the proteins. The combination of three enzymes (trypsin, α-chymotrypsin, and pepsin) had been identified to produce potential anti-inflammatory peptides in microalgae, such as LDAVNR and MMLDF [12]. Table 3 reveals that the use of combined action of two to three enzymes could actually lead to the release of higher numbers of bioactive peptides from the selected proteins. This implies a greater effectiveness of using the combined action of enzymes in cleaving peptide bonds than the single action enzyme, except for pepsin which has almost the same number of released peptides with the combined enzyme action. Pepsin has been reported from several in vitro studies to produce various bioactive peptides from microalgae hydrolysates such as an ACE inhibitor and antioxidant peptides. Samarakoon et al. [9] mentioned that pepsin generated more potential ACE inhibitory peptides compared to other proteases, such as GMNNLTP (IC_50_: 123 μM) and LEQ (IC_50_: 173 μM). Ko et al. [11] also identified LNGDVW from peptic hydrolysates possessed strong scavenged peroxyl, DPPH and hydroxyl radicals at the IC_50_ values of 0.02, 0.92 and 1.42 mM, respectively. Moreover, peptic hydrolysates from microalgae efficiently generated strong antioxidant activities [51,52].

Furthermore, in comparison to the other three proteins, ATP synthase subunit beta demonstrated higher tendency to release more bioactive peptides using the different proteases. However, these theoretically produced bioactive peptides may not always have a comparable function with the in vitro and in vivo analyses, thus further study of these peptides using in vitro and in vivo studies should be conducted. Nevertheless, the BIOPEP’s “enzyme action” tool was able to provide reference information on the possible bioactive peptides that could be released from the selected proteins using various proteases.

## 3. Materials and Methods

### 3.1. Materials

The microalgae, *C. sorokiniana* was obtained from the Taiwan Chlorella Manufacturing Co., Ltd. (Taipei, Taiwan), considered as the largest producer of *Chlorella* every year with an average production of 400 tons of dried biomass [1]. All reagents and chemicals used were analytical grade.

### 3.2. Protein Isolation

The protein isolation process was adapted from the procedure of Parimi et al. [53] with modifications. Briefly, *C. sorokiniana* biomass slurry at 1:16 (*w/v*) ratio was prepared. Sonication for 1 h was done to the slurry for pretreatment and subsequent alkaline protein extraction by solubilization at 11.38 using 2 M NaOH for 35 min with stirring. It was followed by isoelectric precipitation of the supernatant at 4.01 with 1M HCL and stirred for 60 min. Centrifugation at 8750× *g* for 35 min was done for the solid-liquid separation during the solubilization and precipitation steps. The protein isolate was lyophilized and stored at −20 °C until further use. The modified Lowry method [54] was used to determine the protein content of the isolate.

### 3.3. SDS-PAGE Analysis

The SDS-PAGE was performed according to a method described by Schägger and Von Jagow [55] 4% stacking gel (*w/v*) and 12% polyacrylamide gel (w/v). 10 milligrams of protein isolate was dissolved in 1 mL of denaturant sample buffer (0.5 M Tris-HCl pH 6.8, glycerol, 10% SDS, *w/v*, 0.5% bromophenol blue, *w/v*, β-mercaptoethanol), and heated at 95 °C. Then, 10 µL of the sample was loaded to the sample wells. Protein separation was carried at 80 V for 30 min followed by 110 V for 90 min for the resolving gel using a Mini Protean II unit (Bio-Rad Laboratories, Hercules, CA, USA). The gel was stained for 40 min with Brilliant Blue (Bio-Rad, Coomassie R250). Destaining of the gel was done three times using water/methanol/acetic acid (7/2/1, *v/v/v*) for 15 min each cycle with shaking using an orbital shaker (Fristek S10, Taichung city, Taiwan). Estimation of the molecular mass of proteins was done using molecular protein mass marker (250 to 10 kDa, Bio-Rad) loaded at 5 uL in the sample well. The gels was scanned with E-Box VX5 (Vilber Lourmat, Paris, France) and the analysis of the captured image was done using Vision Capt software (V16.08a, Vilber Lourmat, Paris, France).

### 3.4. Proteomics Techniques

#### 3.4.1. In-Gel Tryptic Digestion

The following proteomics technique experiments were carried out in Academia Sinica, Nangang District, Taipei City, Taiwan. Proteomics techniques were adapted from the methods described by Chang et al. [18]. Gel slice and in-gel digestion were performed using the combined modified methods of Rosenfeld et al. [56] and Shevchenko et al. [32]. Briefly, 10 intensive colored protein bands were excised from the SDS-PAGE gel for the in-gel digestion. The gel pieces were destained with 25 mM amonium bicarbonate (ABC)/ 50% acetonitrile (ACN) solution in a microcentrifuge PP tubes. The destained gel pieces were added with 100 µL of 50 mM dithioerythreitol (DTE) / 25 mM ABC and soaked at 37 °C for 1 h. The tubes were centrifuged and the DTE solution was removed. Then, the gel pieces were added with 100 µL of 100 mM iodoacetamide (IAM) / 25 mM ABC and soaked at room temperature in a dark place for 1 h for the alkylation step. The IAM solution was removed after centrifugation. Washing of the gel pieces was done by soaking in 200 µL of 50% ACN / 25 mM ABC for 15 min. The solution was removed after centrifugation and the process was repeated four times. The gel slices were then soaked in 100 µL of 100% ACN for 5 min, repeated twice, and the solution was discarded after centrifugation. The gel slices were dried for 5 min using Speed Vac (Thermo Scientific, Waltham, MA, USA). Trypsin digestion followed by adding Lys-C / 25 mM ABC (enzyme:protein, 1:50) and incubating the mixture for 1 h at 37 °C. Afterwards, the same amount of trypsin was added and incubated for 16 h at 37 °C. Afterwards, the extraction of the tryptic peptides was done with 50 µL of 50% ACN/ 5% trifluoroacetic acid (TFA). The peptide extracts were transferred to new tubes and dried Speed Vac (Thermo Scientific, Waltham, MA, USA). Finally, the peptide extracts were purified using C18 Zip-Tip. The purified peptide extracts were used for the nanoLC–nanoESI MS/MS analysis.

#### 3.4.2. Nanoliquid Chromatography–Nanoelectrospray Ionization Tandem Mass Spectrometry (NanoLC–nanoESI MS/MS) Analysis

Dried tryptic peptide digest was subjected to nanoLC−nanoESI MS/MS analysis using a nanoAcquity system (Waters, Milford, MA, USA) connected to the LTQ Orbitrap Velos hybrid mass spectrometer (Thermo Electron, Bremen, Germany) equipped with a PicoView nanospray interface (New Objective, Woburn, MA). The tryptic peptide mixtures were loaded onto a 75 µm ID, 25 cm length C18 BEH column (Waters, Milford, MA) packed with 1.7 µm particles with a pore width of 130 Å. Separation was performed using a segmented gradient in 60 min from 5 to 35% solvent B (acetonitrile with 0.1% formic acid) at 300 nL/min flow rate and at 35 °C column temperature. Solvent A was 0.1% formic acid in water (*v/v*). The mass spectrometer was operated in the data-dependent mode. In brief, the orbitrap (m/z 350–1600) with the resolution set to 60 K at m/z 400 and automatic gain control (AGC) target at 10^6^ was used to obtain the survey full scan MS spectra. The 20 most intense ions were sequentially isolated for collision-induced dissociation (CID) MS/MS fragmentation and detection in the linear ion trap (AGC target at 10,000) with previously selected ions dynamically excluded for 60 s. Ions with singly and unrecognized charge state were also excluded. The LTQ-Orbitrap data were acquired at the Academia Sinica Common Mass Spectrometry Facilities located at the Institute of Biological Chemistry, Academia Sinica, Nangang District, Taipei City, Taiwan.

#### 3.4.3. Tandem MS Data Analysis of Proteins and Peptide Identification

First, the MS raw data was converted to PKL files using de novo sequencing parameter in the ProteinLynx software coupled with Mascot MS/MS ion search (http://140.112.52.63/mascot/cgi/search_form.pl?FORMVER=2;SEARCH=MIS) [57]. MS/MS data were examined using the National Center for Biotechnology Information (NCBI) database (https://www.ncbi.nlm.nih.gov/) accessed on March 15, 2018 [58] for Viridiplantae (green plants) entries. Search parameters were set to: Carbamidomethyl cysteine as fixed modification; oxidation (M) as variable modification; 10 ppm peptide mass tolerance; 2+, 3+, 4+ peptide charge; ± 0.6 Da MS/MS tolerance; instrument is ESI-TRAP; and the enzyme entry as trypsin with 2 missed cleavages. Peptide masses were acquired as monoisotopic masses.

The Mascot ion score was −10*Log (P), where P is the probability that the observed match is a random event. Individual ion scores of N 45 indicated identity or extensive homology (*p* < 0.05). Protein scores were derived from ion scores as a non-probabilistic basis for ranking protein hits (Matrix Science, London, United Kingdom). The sequence coverage of protein hits was expressed in percentage (%) indicating the sequence homology of identified tryptic peptides from *C. sorokiniana* to corresponding protein hits based on the Mascot MS/MS ion search results [21].

#### 3.4.4. In Silico Analysis of Bioactive Peptides and Enzyme Cleavages using BIOPEP-UWM Database Tools

Sequences of the identified protein of *C. sorokiniana* proteins from NCBI database were analyzed for bioactive peptides and enzyme cleavages using BIOPEP-UWM database (http://www.uwm.edu.pl/biochemia/index.php/pl/biopep) accessed on March 15, 2018 [23] performed as described by Cheung et al. [25] with modifications. Briefly, the bioactivities, sequences, number and location of the peptides were obtained from the sequences of the identified proteins analyzed using the “profiles of potential bioactivity” tool. Moreover, the sequences of the identified proteins were examined using the “enzyme action” tool to simulate enzymatic hydrolysis. A total of 15 enzymatic hydrolysis simulations (composed of 12 individual proteases, one double enzyme action, and two triple enzyme action) were conducted to each protein sequence. A list of all the potential bioactive peptides was obtained after directing the theoretical peptide sequence data to the “search for active fragments” option. The occurrence of the frequency of the bioactive peptides in the intact proteins was computed as A = a/N, where A is occurrence frequency, a is the number of bioactive peptides and N is the total number of amino acid residues in the protein sequence.

## 4. Conclusions

Proteomics techniques coupled with in silico analysis used in this study showed a rapid method to identify the isolated proteins of *C. sorokiniana*, to predict potential bioactivities and to determine the appropriate proteases that theoretically released more bioactive peptides. Results of the proteomics technique showed the identification of tryptic peptides corresponding to eight proteins from the microalgae. The in silico analysis using BIOPEP-UWM database tools revealed that the combined actions of mixed enzymes and the use of single enzyme action of pepsin (pH > 2) could lead to the production of more diverse and larger numbers of potential bioactive peptides embedded in the protein sequences. According to the results, *C. sorokiniana* proteins are potential sources of bioactive peptides with various bioactivities. Nonetheless, with the use of appropriate extraction methods and purification techniques for certain predicted bioactivities, these proteins could be a good alternative source of high value compounds for pharmaceutical, medical, cosmetics and functional food applications to aid in human health maintenance and enhancement.

## Figures and Tables

**Figure 1 ijms-20-01786-f001:**
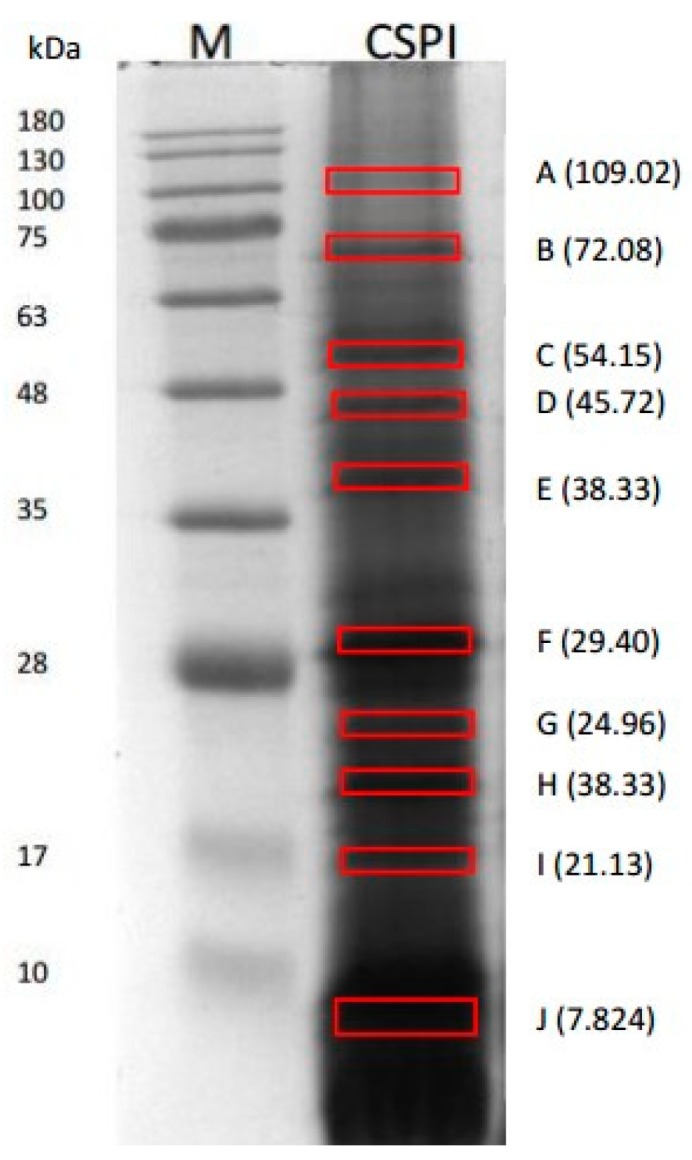
Twelve percent SDS-PAGE of *C. sorokiniana* protein isolates. M: Protein marker; CSPI: *C. sorokiniana* protein isolate.

**Figure 2 ijms-20-01786-f002:**
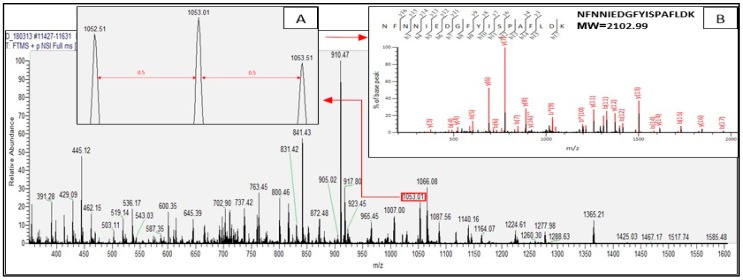
NanoLC–nanoESI MS/MS spectra (m/z region 350–1600) of *C. sorokiniana* protein band D, m/z 1053.401 signal in red box. Insert A presents the identified doubly charged signal by the difference of 0.50 between signals. Insert B illustrates the fragmentation of nanoLC–nanoESI MS/MS spectra of the peptide NFNNIEDGFYISPAFLDK, calculated MW 2102.99 Da.

**Figure 3 ijms-20-01786-f003:**
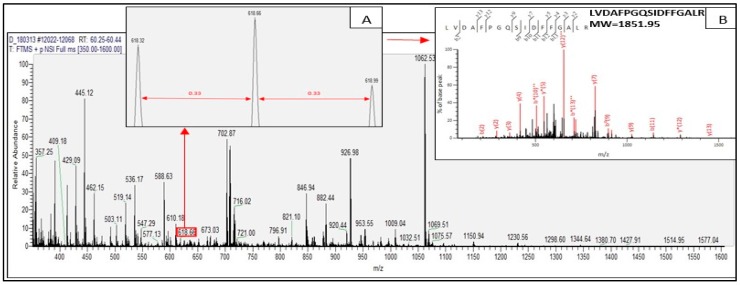
NanoLC–nanoESI MS/MS spectra (m/z region 350–1600) of *C. sorokiniana* protein band D, m/z 618.66 signal in red box. Insert A shows the identified doubly charged signal by the difference of 0.33 between signals. Insert B illustrates the fragmentation of nanoLC–nanoESI MS/MS spectra of the peptide LVDAFPGQSIDFFGALR, calculated MW 1851.95 Da.

**Figure 4 ijms-20-01786-f004:**
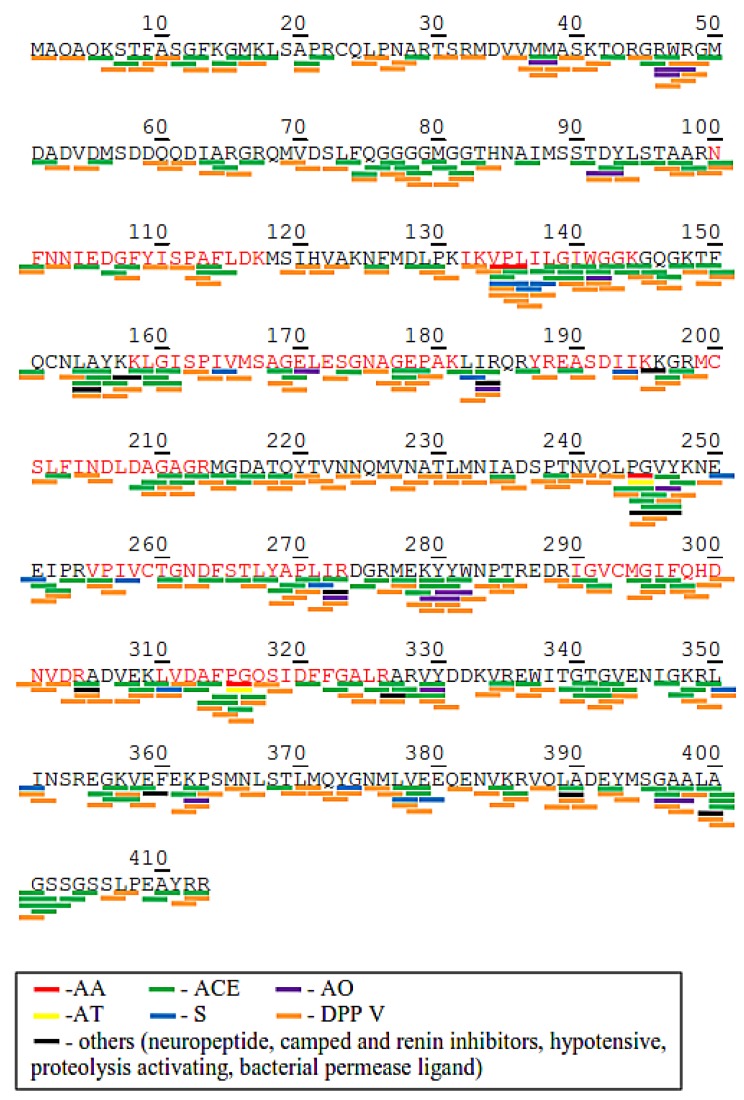
Protein sequence and potential bioactive peptides of chloroplast rubisco activase (AEL29575.1) from *C. sorokiniana*. Abbreviation: ACE Inhibitory (ACE), Antioxidant (AO), Anti-amnestic (AA), Antithrombotic (AT), Stimulating (S), Regulating (R), DPP IV inhibitory (DPP).

**Figure 5 ijms-20-01786-f005:**
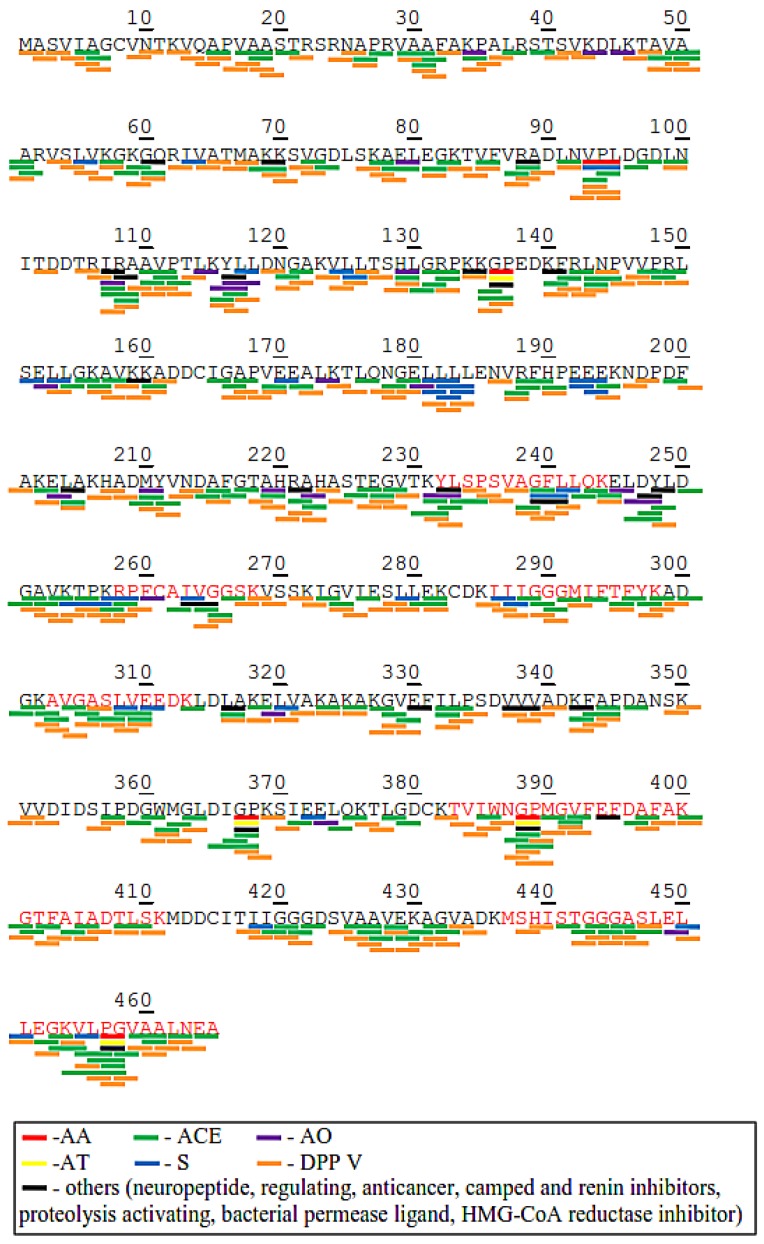
Protein sequence and potential bioactive peptides of phosphoglycerate kinase (AKP17751.1) from *C. sorokiniana*. Abbreviation: ACE Inhibitory (ACE), Antioxidant (AO), Anti-amnestic (AA), Antithrombotic (AT), Stimulating (S), Regulating (R), DPP IV inhibitory (DPP).

**Table 1 ijms-20-01786-t001:** Identified *C. sorokiniana* proteins by SDS-PAGE and nanoLC–nanoESI MS/MS analysis.

Protein Name	Accession Number (NCBI)	Score	Sequence Coverage (%)	Length (AA)	Molecular Weights from NCBI Database (kDa)	Molecular Weights (kDa) Estimated from SDS–PAGE
**Chloroplast Rubisco Activase**	AEL29575.1	323	31%	567	45.68	45.72 (D)
						109.02 (A)
						72.08 (B)
						38.33 (E)
						29.04 (F)
						24.96 (G)
						21.13 (H)
						16.59 (I)
						7.82 (J)
**50S Ribosomal Protein L7/L12 (Chloroplast)**	YP_009020879.1	72	18%	130	13.58	7.82 (J)
**Phosphoglycerate Kinase**	AKP17751.1	250	22%	465	49.13	45.72 (D)
						109.02 (A)
						72.08 (B)
						54.15 (C)
						38.33 (E)
						29.04 (F)
						24.96 (G)
						21.13 (H)
						16.59 (I)
						7.82 (J)
**Fe-superoxide Dismutase**	AHD25899.1	82	8%	236	26.41	24.96 (G)
**Heat Shock Protein 70**	AKP17750.1	128	29%	652	71.55	72.08 (B)
						109.02 (A)
						54.15 (C)
						38.33 (E)
**ATP Synthase Subunit Beta (Chloroplast)**	YP_009020893.1	906	49%	481	51.833	54.15 (C)
**elongation factor 2, partial**	BAE48222.1	120	4%	816	90.54	45.72 (D)
						7.82 (J)
**V-type H+ ATPase Subunit A, partial**	BAE48224.1	61	4%	596	65.47	7.82 (J)

**Table 2 ijms-20-01786-t002:** Number of potential bioactive peptides of identified *C. sorokiniana* proteins using BIOPEP’s “profiles of potential biological activities” tool.

Protein Name	Number of Bioactive Peptides						
	AC	AO	AA	AT	S	R	DPP
**Chloroplast Rubisco Activase**	187 (0.329)	18 (0.032)	3 (0.005)	2 (0.004)	14 (0.25)	3 (0.005)	250 (0.441)
**50s Ribosomal Protein L7/L12 (Chloroplast)**	59 (0.454)	8 (0.062)	1 (0.008)	1 (0.008)	8 (0.062)	1 (0.008)	83 (0.638)
**Phosphoglycerate Kinase**	224 (0.482)	23 (0.049)	5 (0.011)	4 (0.009)	33 (0.071)	6 (0.013)	297 (0.639)
**Fe-Superoxide Dismutase**	107 (0.453)	18 (0.076)	-	-	5 (0.021)	-	156 (0.661)
**Heat Shock Protein 70**	276 (0.423)	30 (0.046)	4 (0.006)	4 (0.006)	41 (0.064)	4 (0.006)	446 (0.684)
**ATP Synthase Subunit Beta (Chloroplast)**	204 (0.424)	20 (0.042)	5 (0.010)	5 (0.010)	25 (0.052)	6 (0.12)	332 (0.690)

Abbreviation: ACE Inhibitory (AC), antioxidant (AO), anti-amnestic (AA), antithrombotic (AT), stimulating (S), regulating (R), dipeptidyl peptidase-IV (DPP IV) inhibitory (DPP).

**Table 3 ijms-20-01786-t003:** Number of predicted potential bioactive peptides to be released from identified proteins of *C. sorokiniana* using BIOPEP’s “enzyme action” tool.

Protease	Chloroplast Rubisco Activase				Phosphoglycerate Kinase				Heat Shock Protein 70				ATP Synthase Subunit Beta			
	AC	AO	S	DP	AC	AO	S	DP	AC	AO	S	DP	AC	AO	S	DP
**Pepsin (EC 3.4.23.1)**	31	2	2	48	29	-	5	51	42	11	1	78	44	6	9	66
**Thermolysin (EC 3.4.24.27)**	12	-	-	14	20	4	-	23	21	-	-	31	20	1	-	25
**Bromelain (EC 3.4.22.32)**	26	3	1	36	25	8	3	41	36	3	2	50	20	2	4	37
**Chymotrypsin A (EC 3.4.21.1)**	5	1	1	10	9	1	2	11	8	2	2	15	7	4	2	10
**Chymotrypsin C (EC 3.4.21.2)**	17	3	1	16	14	-	2	19	20	6	2	35	17	3	4	25
**Pancreatic elastase (EC 3.4.21.36)**	16	2	-	31	27	5	-	49	21	2	-	34	19	2	-	41
**Papain (EC 3.4.22.2)**	31	3	1	35	27	6	2	31	40	4	4	50	27	1	4	43
**Proteinase K (EC 3.4.21.67)**	12	1	-	20	17	1	1	19	23	5	-	41	15	1	-	28
**Trypsin (EC 3.4.21.4)**	5	-	-	3	3	1	-	3	5	-	-	2	3	-	-	3
**Ficin (EC 3.4.22.3)**	27	7	2	37	22	8	3	33	26	4	3	35	22	7	1	28
**Cathepsin (EC 3.4.21.20)**	4	1	1	6	7	1	1	6	6	2	1	7	5	3	1	6
**Subtisilin (EC 3.4.21.62)**	10	4	2	14	10	3	4	18	9	5	7	24	12	2	6	23
**Pepsin + Trypsin**	28	4	2	55	33	1	5	56	36	2	10	66	40	6	9	62
**Pepsin + Trysin + Chymotrypsin A**	31	4	1	40	33	1	5	54	34	1	10	63	38	5	9	61
**Pepsin + Trysin + Chymotrypsin C**	28	4	1	33	35	1	4	54	41	2	10	70	38	5	9	51

Abbreviation: ACE Inhibitory (AC), Antioxidant (AO), Stimulating (S), DPP IV inhibitory (DP).

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
