# Peer review of "Prediction of Bioactive Peptides from Chlorella sorokiniana Proteins Using Proteomic Techniques in Combination with Bioinformatics Analyses"

_ijms, 2019, doi:10.3390/ijms20071786_

Reviewer 1 Report

In this paper, Authors presented a procedure of protein isolation from Chlorella sorokiniana. Isolated proteins were then subjected to in silico analysis to predict potential peptides with biological activities. They identified a total of 8 proteins by proteomics techniques and found these proteins have highest number of dipeptidyl peptidase‐IV (DPP IV) inhibitors, with high occurrence of other bioactive peptides, such as angiotensin‐I converting enzyme (ACE) inhibitors, glucose uptake stimulants, antioxidants, regulating peptides, antiamnestic peptides and antithrombotic peptides.

I found this paper interesting and scientifically sound. However, there are some major concerns that should be addressed by the Authors before it can be reconsidered for publication.

Introduction:

- The first paragraph of the Introduction should be rewritten to make the reading more fluent. In particular, when citing the different previously published papers, Authors should avoid to do this by making a sort of list of who has reported what, rather they should better link the most relevant findings of these studies.

- In the Introduction, Authors mainly referred to the proteic extracts of Chlorella sorokiniana. However, a description of other compounds which can be obtained from this microalga, such as such as omega (ω)-3 and ω-6 polyunsatured fatty acids (PUFAs), and of the in vivo effects of these extracts (Morgese et al., Molecules. 2016 Sep 29;21(10). pii: E1311) should be added in order to have a complete description of the extraction possibilities.

Results:

- The graphical aspect of Figure 1 should be significantly improved, as in its actual state it appears quite difficult to see what indicated by the Authors in its description.

- Figures 4 and 5: in their legends, Authors indicated that ACE is abbreviated as AC. However, in the Figure ACE is still indicated. Moreover, the used colours should be changed, as it is difficult to distinguish between similar colours (i.e. yellow and light orange).

Materials and Methods:

References for “NanoLiquid chromatography–nanoelectrospray ionization tandem mass spectrometry (NanoLC–nanoESI–MS/MS) analysis” should be included.

Conclusions:

This section of the manuscript should be further elaborated, better highlighting the impact of the findings reported by the Authors in the field of microalgae extract use.

Author Response

Response to Reviewer 1 Comments

We thank Reviewer 1 for his/her overall appreciation of our manuscript. His/her comments were addressed accordingly:

In this paper, Authors presented a procedure of protein isolation from Chlorella sorokiniana. Isolated proteins were then subjected to in silico analysis to predict potential peptides with biological activities. They identified a total of 8 proteins by proteomics techniques and found these proteins have highest number of dipeptidyl peptidase‐IV (DPP IV) inhibitors, with high occurrence of other bioactive peptides, such as angiotensin‐I converting enzyme (ACE) inhibitors, glucose uptake stimulants, antioxidants, regulating peptides, antiamnestic peptides and antithrombotic peptides.

I found this paper interesting and scientifically sound. However, there are some major concerns that should be addressed by the Authors before it can be reconsidered for publication.

Introduction:

Point 1: The first paragraph of the Introduction should be rewritten to make the reading more fluent. In particular, when citing the different previously published papers, Authors should avoid to do this by making a sort of list of who has reported what, rather they should better link the most relevant findings of these studies.

Response 1: Corrections have been made. Please refer to Line 35 -48.

Point 2: In the Introduction, Authors mainly referred to the proteic extracts of Chlorella sorokiniana. However, a description of other compounds which can be obtained from this microalga, such as such as omega (ω)-3 and ω-6 polyunsatured fatty acids (PUFAs), and of the in vivo effects of these extracts (Morgese et al., Molecules. 2016 Sep 29;21(10). pii: E1311) should be added in order to have a complete description of the extraction possibilities.

Response 2: The additional description and reference have been added in Line 59-60.

Results:

Point 3: The graphical aspect of Figure 1 should be significantly improved, as in its actual state it appears quite difficult to see what indicated by the Authors in its description.

Response 3: As per advice, necessary improvement was made as shown in Figure 1.

Point 4: Figures 4 and 5: in their legends, Authors indicated that ACE is abbreviated as AC. However, in the Figure ACE is still indicated. Moreover, the used colours should be changed, as it is difficult to distinguish between similar colours (i.e. yellow and light orange).

Response 4: Corrections have been made.

Materials and Methods:

Point 5: References for “NanoLiquid chromatography–nanoelectrospray ionization tandem mass spectrometry (NanoLC–nanoESI–MS/MS) analysis” should be included.

Response 5: The protocol used for NanoLiquid chromatography–nanoelectrospray ionization tandem mass spectrometry (NanoLC–nanoESI–MS/MS) analysis followed the mandatory protocol of the research center where author used the machine.

Conclusions:

Point 6: This section of the manuscript should be further elaborated, better highlighting the impact of the findings reported by the Authors in the field of microalgae extract use.

Response 6: Conclusions have been revised.

Reviewer 2 Report

In this manuscript, Tejano et. al. use proteomics and in silico analysis to study proteins that are isolated from Chlorella Sorokiniana. Using the in silico analytical tools, the authors predict the potential biological activities of the peptides that are produced by enzymatic hydrolysis simulations.  The authors results suggest that C. Sorokiniana is a good source of bioactive peptides that can replace existing high-value compounds for a variety of applications. The results are clearly presented and overall, the work is well done. 

Some minor suggestions for improvements include: 

1.  The authors should avoid the use of acronyms in titles (BIOPEP-UWM). Suggest replacing "BIOPEP-UWM" in the manuscript title with "bioinformatics". 

 2. Please define  allacronyms prior to first use. For e.g. in the introduction,  acronyms like ACE or DDP-IV need to be defined. This makes the manuscript readable to audiences with diverse backgrounds. 

3. The introduction should include references to prior work done on identifying bioactive peptides from C. Sorokiniana.  For e.g. https://www.mdpi.com/2072-6643/10/10/1397 or https://www.mdpi.com/1660-3397/13/10/6152/pdf 

4. The lack of a discussion of the results obtained by the authors in the context of results from others working towards identifying bioactive peptides from microalgae is a bit jarring and needs to be included in section 4.

5. The conclusions need to layout the next steps in ensuring these microalgae can be further developed as alternative sources for high-value compounds with a variety of applications. 

Author Response

Response to Reviewer 2 Comments

We are grateful to the kind appreciation of our submitted manuscript. His/her comments were addressed below:

In this manuscript, Tejano et. al. use proteomics and in silico analysis to study proteins that are isolated from Chlorella Sorokiniana. Using the in silico analytical tools, the authors predict the potential biological activities of the peptides that are produced by enzymatic hydrolysis simulations.  The authors results suggest that C. Sorokiniana is a good source of bioactive peptides that can replace existing high-value compounds for a variety of applications. The results are clearly presented and overall, the work is well done. 

Some minor suggestions for improvements include: 

Point 1: The authors should avoid the use of acronyms in titles (BIOPEP-UWM). Suggest replacing "BIOPEP-UWM" in the manuscript title with "bioinformatics". 

Response 1: Suggestion has been made. “Prediction of bioactive peptides from Chlorella sorokiniana proteins using proteomic techniques in combination with bioinformatics”

Point 2: Please define all acronyms prior to first use. For e.g. in the introduction, acronyms like ACE or DDP-IV need to be defined. This makes the manuscript readable to audiences with diverse backgrounds. 

Response 2: Corrections have been made. Please refer to Line 53 and 64

Point 3: The introduction should include references to prior work done on identifying bioactive peptides from C. Sorokiniana.  For e.g. https://www.mdpi.com/2072-6643/10/10/1397 or https://www.mdpi.com/1660-3397/13/10/6152/pdf

Response 3: Suggestion has been added in Line 56-62.

Point 4: The lack of a discussion of the results obtained by the authors in the context of results from others working towards identifying bioactive peptides from microalgae is a bit jarring and needs to be included in section 4.

Response 4: Suggestion has been carried out in Section 2 as “Results and discussion” section in this paper to make the reading more fluent.

Point 5: The conclusions need to layout the next steps in ensuring these microalgae can be further developed as alternative sources for high-value compounds with a variety of applications. 

Response 5: Corrections have been made in the conclusions section.

Reviewer 3 Report

The authors of this paper have carried out the following analyses:

(1) They extracted proteins from Chlorella sorokiniana using SDS lysis buffer and ran an SDS PAGE gel of the resultant extract

(2) Using LCMS they identified proteins from 10 major bands from the gel lane

(3) From the NCBI sequences of 6 of the identified proteins, they used the BIOPEP database to deduce theoretical peptides produced by digestion with enzymes to produce lists of theoretical peptides. They focus on pepsin, bromelain & papain digestion.

(4) They conclude that Chlorella sorokiniana is a promising source of bioactive peptides.

The major problem with this manuscript is the lack of a substantial contribution to the field, as well as various conceptual errors. 

First the authors conclude that the identified proteins correspond to the 10 major bands on their SDS-PAGE gel.  This has not been proven.  Clearly the identified proteins are present in the extract (it would be surprising if they were not, as they are all Chlorella proteins, and all likely to be relatively abundant). However, they have not addressed the possibility that these proteins might be less abundant than whatever protein is producing the major band they identify.  The major proteins by weight in algae are ribosomes and the photosynthetic proteins (Rubisco, PSI, PSII, the LHC proteins, cytochrome b6f and the ATPase).  Of these, only 3 have been identified by the authors. Where are the others? Because most are membrane proteins, they yield fewer tryptic peptides than other proteins and are therefore less likely to be detected – but this doesn't mean they are not the major proteins by weight, so the author's conclusion that their identifications are the major Chlorella proteins is unsubstantiated. This is important because protease digestion will yield only small amounts of peptides from proteins that are NOT the major components.

Second, why carry out the whole SDS-PAGE and LCMS exercise in the first place? Abundant evidence in the literature already identifies the major proteins of plants and algae, and since the Chlorella genome has been sequenced, all these proteins are already avaliable in the databases.  There is no need to do LCMS analysis to identify their sequences.  However well carried out, the LCMS analysis yields no new knowledge.

Third, I view the BIOPEP analysis as the start of the analysis, not its end. Having concluded from a relatively brief interrogation of the database that a set of proteins such as these have the potential to yield many di- and tripeptides with theoretical activity, there are two logical paths forward.  One is to do an extensive in silico analysis, perhaps comparing Chlorella with other potential sources (e.g. milk peptides) or other algae for example, to try to conclude something new and non-intuitive from the analysis (for example, that maybe Chlorella has a high activity peptide not present in other algae, or alternatively, that all algae are much the same in terms of their bioactive peptide content).  The other is to actually do some experiments to prove that these theoretical peptides can actually be generated and show activity in the real world, instead of just in silico (to review the clear and obvious limitations of in silico digestion, I refer the authors to the 2018 review by Agyei et al. in Anal Bioanal Chem 410:3463).

As examples of substantial bioinformatic analyses I offer Zhu et al. (2017) Eur Food Res Technol 243:1739 (though even here the authors synthesise a peptide and do some biochemical analysis) while Yu et al. (2018) in Food Func 9:364  use BIOPEP to identify a theoretical ACE inhibitor peptide and then synthesise it and assay it. 

In order to be considered for publication the current manuscript would need to offer some kind of new insight into the potential for Chlorella to generate bioactive peptides, such as actual data, or a comprehensive in silico analysis that represents a substantial amount of work and which yields non-obvious value to the reader.

Author Response

Response to Reviewer 3 Comments

We are pleased for your kind consideration and insightful comments on our manuscript. Comments were taken positively and hopefully addressed with our answers below:

The authors of this paper have carried out the following analyses:

(1) They extracted proteins from Chlorella sorokiniana using SDS lysis buffer and ran an SDS PAGE gel of the resultant extract

(2) Using LCMS they identified proteins from 10 major bands from the gel lane

(3) From the NCBI sequences of 6 of the identified proteins, they used the BIOPEP database to deduce theoretical peptides produced by digestion with enzymes to produce lists of theoretical peptides. They focus on pepsin, bromelain & papain digestion.

(4) They conclude that Chlorella sorokiniana is a promising source of bioactive peptides.

Point 1: First the authors conclude that the identified proteins correspond to the 10 major bands on their SDS-PAGE gel.  This has not been proven.  Clearly the identified proteins are present in the extract (it would be surprising if they were not, as they are all Chlorella proteins, and all likely to be relatively abundant). However, they have not addressed the possibility that these proteins might be less abundant than whatever protein is producing the major band they identify.  The major proteins by weight in algae are ribosomes and the photosynthetic proteins (Rubisco, PSI, PSII, the LHC proteins, cytochrome b6f and the ATPase).  Of these, only 3 have been identified by the authors. Where are the others? Because most are membrane proteins, they yield fewer tryptic peptides than other proteins and are therefore less likely to be detected – but this doesn't mean they are not the major proteins by weight, so the author's conclusion that their identifications are the major Chlorella proteins is unsubstantiated. This is important because protease digestion will yield only small amounts of peptides from proteins that are NOT the major components.

Response 1: The proteins mentioned in the paper were the identified proteins from excised bands in the SDS PAGE gel.  We acknowledge the possibility of some proteins not detected from the sample as per iteration that the protein extraction method could not have extracted most of the major proteins in the sample. However, we have not concluded that the identified proteins were the major proteins present but we presented that from what we did, these were the relatively major proteins identified based on the results.

Point 2: Second, why carry out the whole SDS-PAGE and LCMS exercise in the first place? Abundant evidence in the literature already identifies the major proteins of plants and algae, and since the Chlorella genome has been sequenced, all these proteins are already avaliable in the databases.  There is no need to do LCMS analysis to identify their sequences.  However well carried out, the LCMS analysis yields no new knowledge.

Response 2: Although, there are already available information on the Chlorella genome identifying their major proteins, we still decided to conduct the LC/MS analysis to further note the proteins that could found specific to our sample. The LC/MS results also gave us a relatively solid justification why certain proteins were further used in the bioinformatics analyses. We thought that by just relying on the databases, it will be harder for us to decide what major proteins we should use. With the LC/MS results, we have date to support why we used those proteins for further analyses.

Point 3: Third, I view the BIOPEP analysis as the start of the analysis, not its end. Having concluded from a relatively brief interrogation of the database that a set of proteins such as these have the potential to yield many di- and tripeptides with theoretical activity, there are two logical paths forward.  One is to do an extensive in silico analysis, perhaps comparing Chlorella with other potential sources (e.g. milk peptides) or other algae for example, to try to conclude something new and non-intuitive from the analysis (for example, that maybe Chlorella has a high activity peptide not present in other algae, or alternatively, that all algae are much the same in terms of their bioactive peptide content).  The other is to actually do some experiments to prove that these theoretical peptides can actually be generated and show activity in the real world, instead of just in silico (to review the clear and obvious limitations of in silico digestion, I refer the authors to the 2018 review by Agyei et al. in Anal Bioanal Chem 410:3463).

Response 3: We also acknowledge that the in silico analysis is not the end but the start of the analysis, mainly if not solely, the basis for conducting sound and logical in vitro and in vivo analyses. One of the main take-away from this paper, is the knowledge that this microalgae is a potential source of bioactive peptides and in order to eventually benefit from it, further in vitro and in vivo studies should be done. The concern is hopefully addressed in the conclusion of the paper, wherein suggestion has been made to further the study. Please refer to lines 380-387.

Page 15, Line 380-387: Results suggest the potential of C. sorokiniana proteins as source of bioactive peptides such as DPP IV inhibitors and ACE inhibitors, among others. In-vitro and in-vivo analyses can be done to verify the beneficial effects of the bioactive compounds revealed by this in silico study. Nonetheless, with the use of appropriate extraction methods and purification techniques, these proteins can be a good alternative source of high value compounds for pharmaceutical, medical, cosmetics, and functional food applications to aid in human health maintenance and enhancement.

Round  2

Reviewer 1 Report

I am satisfied with the answers provided by the Authors to my comments.

Author Response

Thank you for giving us insightful comments.